# Approximate Bayesian Inference via Bitstring Representations

**Aleksanteri Sladek**[1]         **Martin Trapp**[1]         **Arno Solin**[1]

[1]Department of Computer Science, Aalto University, Espoo, Finland

## Abstract

The machine learning community has recently put effort into quantized or low-precision arithmetics to scale large models. This paper proposes performing probabilistic inference in the quantized, discrete parameter space created by these representations, effectively enabling us to learn a continuous distribution using discrete parameters. We consider both 2D densities and quantized neural networks, where we introduce a tractable learning approach using probabilistic circuits. This method offers a scalable solution to manage complex distributions and provides clear insights into model behavior. We validate our approach with various models, demonstrating inference efficiency without sacrificing accuracy. This work advances scalable, interpretable machine learning by utilizing discrete approximations for probabilistic computations.

## 1 INTRODUCTION

Probabilistic inference is central to modern machine learning, providing a principled framework for reasoning under uncertainty. In Bayesian inference, uncertainty is captured through probability distributions over parameters, with Bayes' theorem offering a systematic way to update beliefs with data. However, exact Bayesian inference is often intractable due to the complexity of the integrals involved. Variational inference (VI) [Blei et al., 2017, Jordan et al., 1999, Wainwright and Jordan, 2008] is typically employed as a scalable alternative to Markov chain Monte Carlo (MCMC) methods, enabling inference in high-dimensional models. Despite its success, VI relies on continuous parameterizations and often restrictive Gaussian assumptions, which can introduce representational and computational inefficiencies, particularly in large-scale settings.

To address computational constraints, the machine learning

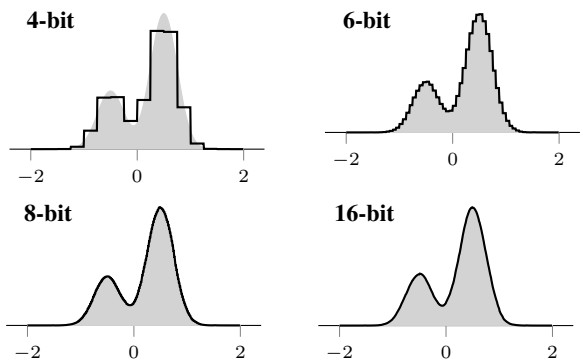

Figure 1: **Capturing a 1D Gaussian mixture** with BitVI with different numbers of bits in the bitstring. Even the 4-bit result serves a practical purpose, while the model saturates around 8 bits when compared to its 16 bit version.

community has increasingly embraced quantization techniques. These methods reduce numerical precision to improve efficiency, leveraging low-bit representations for storage and computation. Many of those can be related to reducing the numerical precision, such as developing tailored low-precision number systems [Gustafson and Yonemoto, 2017, Agrawal et al., 2019] or methods for parameter quantization. Recent works leveraging large-scale mixed-precision FP8 [*e.g.*, Liu et al., 2024], FP4 [Wang et al., 2025], or even 1-bit neural architectures [Ma et al., 2024] have shown innovative low-precision training approaches.

Fig. 1 illustrates how a Gaussian mixture model, typically represented in high-precision floating point, can be equivalently expressed using a low-precision bitstring representation, motivating the feasibility of inference in quantized spaces. These developments suggest that probabilistic inference need not be tied to continuous-valued computations but can instead be formulated in the space of bitstrings.

This work hinges on the fundamental principle that on a computer, continuous values are represented by finite-length bitstrings—that is, a discrete representation. Hence, prob-

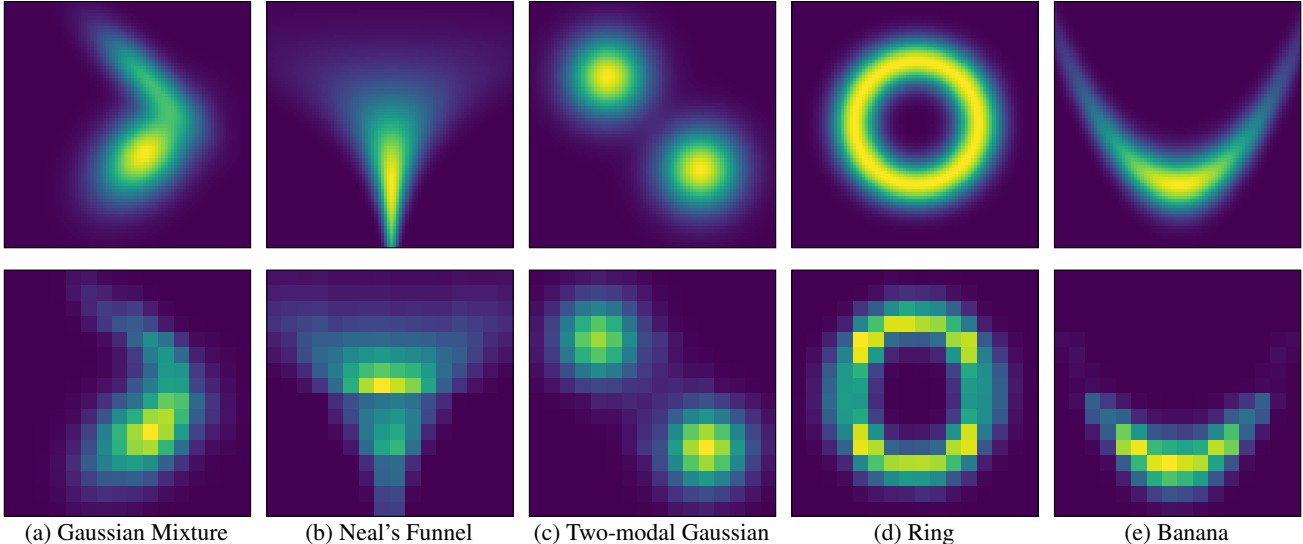

Figure 2: Exact target densities and BitVI (4-bit) on non-Gaussian 2D density functions. We capture the overall and cross-densities well despite the low bit precision. We include comparisons to full-covariance VI in Fig. 11.

ability distributions representable with a computer necessarily possess a representation over finite-length bitstrings. We explore this connection and derive a method to conduct efficient approximate inference through this link.

This work introduces **BitVI**, a novel approach for approximate probabilistic inference in bitstring models. BitVI exploits the inherent discrete nature of number representations to approximate continuous distributions directly in the space of bitstrings. By leveraging probabilistic circuits [Darwiche, 2003, Choi et al., 2020], our method provides a tractable way to learn and perform inference over complex distributions without requiring high-precision representations. Fig. 2 demonstrates how BitVI can model complex distribution with only 4-bit precision.

We validate BitVI across *(i)* standard benchmark densities, demonstrating its ability to approximate known distributions; and *(ii)* Bayesian deep learning in neural network models in *Bayesian Benchmarks*, where BitVI enables scalable and direct uncertainty quantification. Our results highlight the efficiency and accuracy of BitVI, making it a compelling alternative to traditional inference methods.

Our contributions can be summarized as follows.

- **Methodological:** We introduce BitVI, a novel approach for approximate Bayesian inference in bitstring models, leveraging probabilistic circuits for efficient learning and inference.

- **Experimental:** We provide proof-of-concept and benchmarking results on standard test problems as well as Bayesian deep learning tasks, demonstrating the effectiveness of BitVI in practical applications.

- **Insights:** We explore the role of bitstring representa-

tions in probabilistic inference and shed light on the trade-offs between model flexibility and quantization.

## 2 BACKGROUND AND RELATED WORK

The relationship between continuous and discrete representations is fundamental to computational science. At its core, digital computation relies on discrete structures, with real-valued quantities encoded as finite-length bitstrings [Ch. 4 Knuth, 1997]. Floating-point arithmetic provides an approximation to continuous values within this discrete framework, ensuring efficient numerical operations while introducing inherent precision limitations [Ch. 1 Sterbenz, 1974]. In recent years, this foundational connection has gained renewed attention in machine learning, particularly due to advances in quantization and low-precision arithmetic. While these techniques are primarily motivated by hardware constraints, they also present an opportunity: if inference can be formulated directly over discrete bitstring representations, it may unlock new efficiencies in probabilistic modeling.

Bayesian inference provides a principled framework for reasoning under uncertainty, yet exact inference remains intractable in most real-world scenarios. This has led to the development of approximate inference techniques, such as variational inference (VI) [Blei et al., 2017, Jordan et al., 1999, Wainwright and Jordan, 2008]. VI formulates inference as an optimization problem, where a parametric distribution is fitted to approximate the posterior while minimizing the reverse KL divergence. Despite its scalability, VI is often constrained by its reliance on continuous parameterizations, which can introduce numerical instabilities and bias due to restrictive approximations, *e.g.*,

$$\hat{q}(b_1, b_2, b_3) = f\left( \vphantom{\Big|} \right) \xrightarrow{\text{induces}} q(x) = (f \circ \phi^{-1})\left( \vphantom{\Big|} \right)$$

Distribution over bitstrings $\boldsymbol{b}$        Distribution over fixed-point numbers $x$

Figure 3: **Illustration of our method:** For the case of fixed-point numbers, where we use the bitstring to up to each sum node to index the sum in the circuit. The bitstring can be visualized as a hypercube, and the PC induces a distribution over the fixed-point numbers represented by the bitstring.

mean-field or unimodality assumptions. These limitations are apparent when operating under low-precision, raising the question: *Can we perform inference directly in a discrete representation space?*

Probabilistic circuits (PCs) are a recent framework to study tractable representations of complex probability distributions [Choi et al., 2020]. Depending on the structural properties of the PC, certain inference scenarios can be rendered tractable (polynomial in the model complexity) under the circuit while maintaining a high expressivity. While PCs are typically employed for exact probabilistic inference, they have found successful application in approximate Bayesian inference, for example, as surrogate through compilation [Lowd and Domingos, 2010], as variational distribution for structured discrete models [Shih and Ermon, 2020], or in discrete probabilistic programs [Saad et al., 2021]. Our work is closely related to work by Garg et al. [2024], which utilized PCs over bitstring representation for efficient approximate inference in probabilistic programs. This work highlights that PCs are a natural and promising representational framework for approximate Bayesian and uncertainty quantification.

## 3 METHODS

Given a target density $p$, we aim to find a variational approximation $q$ that minimizes the divergence of $p$ from $q$. As commonly done, we will focus on the reverse Kullback–Leibler (KL) divergence of $q$ from $p$, instead of the forward KL. Moreover, we assume that $q$ takes a parametric form with parameters $\boldsymbol{\theta}$, *i.e.*, $q_{\boldsymbol{\theta}}$. Thus, the goal is to find $\boldsymbol{\theta}$ such that

$$\text{KL}(q_{\boldsymbol{\theta}} \,\|\, p) = \int_{x \in \mathcal{X}} q_{\boldsymbol{\theta}}(x) \log\left( \frac{q_{\boldsymbol{\theta}}(x)}{p(x)} \right) \mathrm{d}x, \quad (1)$$

is minimized, assuming that $\mathcal{X} \subseteq \mathbb{R}^d$ for some $d \geq 1$.

In general, computing Eq. (1) is intractable for two reasons: *(i)* $p$ is often only known up to an unknown normalization constant $Z_p$ and *(ii)* $p$ and $q$ do not exhibit sufficient

structure to render the integration tractable [Wang et al., 2024]. Henceforth, one typically optimizes the evidence lower bound (ELBO), which can be written as

$$\mathcal{L}(q_{\boldsymbol{\theta}}, p) = \mathbb{E}_{x \sim q_{\boldsymbol{\theta}}} [\log p(x)] + \mathcal{H}(q_{\boldsymbol{\theta}}), \quad (2)$$

where $\mathcal{H}(q_{\boldsymbol{\theta}}) = -\mathbb{E}_{x \sim q_{\boldsymbol{\theta}}} [\log q_{\boldsymbol{\theta}}(x)]$ denotes the entropy of the variational distribution $q_{\boldsymbol{\theta}}$. In case $q_{\boldsymbol{\theta}}$ admits a tractable entropy computation, only the first term in Eq. (2) requires numerical approximation.

Crucially, when computing either Eq. (1) or Eq. (2) on a computer, each $x$ will inevitably be represented in a discretized form. In fact, every real-valued number is represented by a series of bitstrings and mapped to the real line by a mapping function $\phi \colon \{0, 1\}^B \to \mathbb{R}$ given by the chosen number system. Consequently, any distribution $p$ or $q$ represented on a computer can be expressed in terms of a distribution over bitstrings. Fig. 4 illustrates the representation of a real-valued number using an 8-bit fixed-point representation.

$$-2.375 = \boxed{1}\,\boxed{0}\,\boxed{1}\,\boxed{0}\,\boxed{0}\,\boxed{1}\,\boxed{1}\,\boxed{1} \quad \text{(8-bit fixed-point)}$$
$$\underbrace{\phantom{1}}_{\text{sign}}\,\underbrace{\phantom{010}}_{\text{integer}}\,\underbrace{\phantom{0111}}_{\text{fraction}}$$

Figure 4: Representation of '$-2.375$' using an 8-bit fixed-point number system with sign, integer, and fraction bits.

In the following, we will exploit that continuous distributions can be represented by defining a distribution over bitstrings to formulate a tractable and flexible variational family.

### 3.1 BITVI: VARIATIONAL DISTRIBUTIONS OVER BITSTRING REPRESENTATIONS

Let $\hat{q}$ be a distribution over binary strings with probability measure $\hat{Q}$ defined on the measurable space of binary strings $(\mathcal{Y}, \mathcal{A})$ with corresponding $\sigma$-algebra $\mathcal{A}$. Further, let

$(\mathbb{R}, \mathcal{B})$ be the measurable space of real numbers with Borel $\sigma$-algebra $\mathcal{B}$. Define a measurable mapping $\phi \colon \mathcal{Y} \to \mathbb{R}$ that assigns to each binary string a real number according to a specified number system, for example, the fixed point representation. The induced probability measure $Q$ on $(\mathbb{R}, \mathcal{B})$ is the pushforward measure of $\hat{Q}$ through $\phi$. Specifically, for any Borel set $B \in \mathcal{B}$ we have $Q(B) = \hat{Q}(\phi^{-1}(B))$ where $\phi^{-1}(B)$ is the pre-image of $B$ under $\phi$. Finally, we represent the density $q$ of $Q$ using a (deterministic) probabilistic circuit (PC). The resulting construction is illustrated in Fig. 3 for the case of fixed-point numbers, where we use the bitstring up to each sum node to index the sum in the circuit. Note that for fixed-point representations with infinite precision, this construction is equivalent to probability measures generated by Pólya trees [Ferguson, 1974, Trapp and Solin, 2022].

**Definition 3.1** (Deterministic Probabilistic Circuit). *A probabilistic circuit $f(\boldsymbol{x})$ is a multi-linear function represented by a computational graph consisting of sum nodes $\mathsf{S}(\boldsymbol{x}) = \oplus_i w_i f_i(\boldsymbol{x})$, product nodes $\mathsf{P}(\boldsymbol{x}) = \otimes_i f_i(\boldsymbol{x})$, and leaf nodes consisting of tractable (univariate) functions $\psi_i(x)$. The circuit $f$ characterizes a multivariate probability distribution over random variables $X_1, \dots, X_d$ by, for example, representing its mass, density, or characteristic function [Yu et al., 2023, Broadrick et al., 2024]. Note that we assume that the circuit is smooth and decomposable [Choi et al., 2020] and refer to Appendix A for details.*

*We call a sum node $\mathsf{S}$ deterministic if for each $\boldsymbol{x}$, only one summand is positive. Consequently, $f$ is deterministic if all sum nodes are deterministic [Choi et al., 2020].*

By specifying a $\hat{q}$ over bitstrings and a respective number system, we obtain an induced variational distribution $q$ on the real line. As previously mentioned, our goal is to find a parameterization $\boldsymbol{\theta}$ of our variational distribution such that Eq. (1) is minimal. When representing $q$ using a deterministic PC, the parameters $\boldsymbol{\theta}$ correspond to the collection of weights $\{w_i\}_i$ of the circuit. Note that by construction, the leaf nodes of our circuit model are continuous uniform distributions and, therefore, do not have any additional parameters. The resulting deterministic PC is a tree with depth proportional to the number of bits used in the bitstring representation. Each sum node in the PC represents the decision of a bit and weights correspond to the conditional probability of the respective decision. For example, the probability of 0.5 in 3-bit fixed-point number system with one integer bit and no sign-bit, which corresponds to the bitstring 010, is computed by obtaining the bit decisions, *i.e.*, $b_0 = 0$, $b_1 = 1$, and $b_2 = 0$, and evaluating the circuit along the respective path, *i.e.*, $p(x = 0.5) = w_0 w_{01} w_{010} \frac{1}{2^{B_{\text{frac}}}}$ where $B_{\text{frac}} = 2$ is the number of fraction bits. Fig. 5 illustrates the decision process represented by the circuit.

**Depth Regularization** To encourage that $q$ has a smooth density in the limit of infinite precision, we leverage a depth

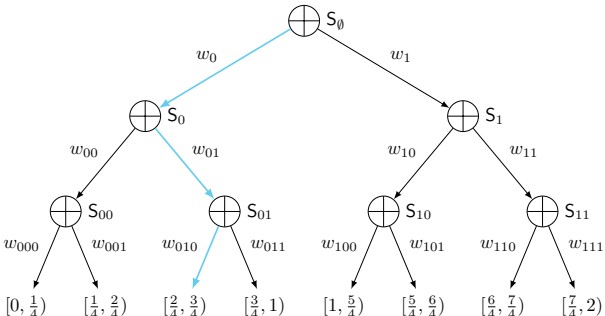

Figure 5: The decision process represented by the circuit.

regularization. The depth regularization is based on Pólya tree prior constructions for priors over continuous probability distributions. Specifically, Ferguson [1974] proposed to use a Beta prior on each weight of a Pólya tree with symmetric $\alpha$-parameter that has a quadratic increase in depth $j$ of the tree, *i.e.*, $\alpha(j) = j^2$. An alternative parameterization is given by Castillo [2017] as $\alpha(j) = 2^j$. In essence, both approaches ensure that the prior probability of uniformly distributed weights increases with depth. We adopt this approach and use Laplace smoothing of the circuit weights with a depth-dependent smoothing factor. In particular, for bit $b_j$ (depth $j$) with $j \geq 0$ we define each weight for $b_j = 0$ as

$$w_{\epsilon 0} = \frac{v_{\epsilon 0} + c\alpha(j)}{v_{\epsilon 0} + v_{\epsilon 1} + 2c\alpha(j)}, \quad (3)$$

where $\epsilon$ denotes a $j - 1$ long binary string, $v_{\epsilon 0} > 0$ is an unnormalized weight, and $c > 0$ is a hyperparameter. The weight for $\epsilon 1$ is given analogously.

**Computation of the ELBO** A particular property of deterministic PCs is that the entropy can be computed in linear time w.r.t. the number of edges of the circuit [Vergari et al., 2021] (see Appendix B for details). As such, we only need to approximate the expected log probability in Eq. (2) using Monte Carlo (MC) integration. To do so, we first use a reparameterization using the inverse CDF transform, which is available analytically in the case of deterministic PCs.

In particular, we reparameterize the ELBO as,

$$\mathcal{L}(q_{\boldsymbol{\theta}}, p) = \mathbb{E}_{u \sim \mathsf{Unif}(0,1)} \left[ \log p(F_{q_{\boldsymbol{\theta}}}^{-1}(u)) \right] + \mathcal{H}(q_{\boldsymbol{\theta}}), \quad (4)$$

where $F_{q_{\boldsymbol{\theta}}}^{-1}(\cdot)$ is the inverse CDF transform of $q_{\boldsymbol{\theta}}$. We then generate $T$ samples from a uniform distribution $u^s \sim \mathsf{Unif}(0, 1)$ and compute a MC estimate of Eq. (4), *i.e.*,

$$\mathcal{L}(q_{\boldsymbol{\theta}}, p) \approx \frac{1}{T} \sum_{s=1}^{S} \log p(F_{q_{\boldsymbol{\theta}}}^{-1}(u_s)) + \mathcal{H}(q_{\boldsymbol{\theta}}). \quad (5)$$

Note that Eq. (5) can be computed efficiently.

**Remark 3.2.** *The inverse CDF transform of $q_{\boldsymbol{\theta}}$ can be computed in linear time w.r.t. the depth of the circuit.* ◁

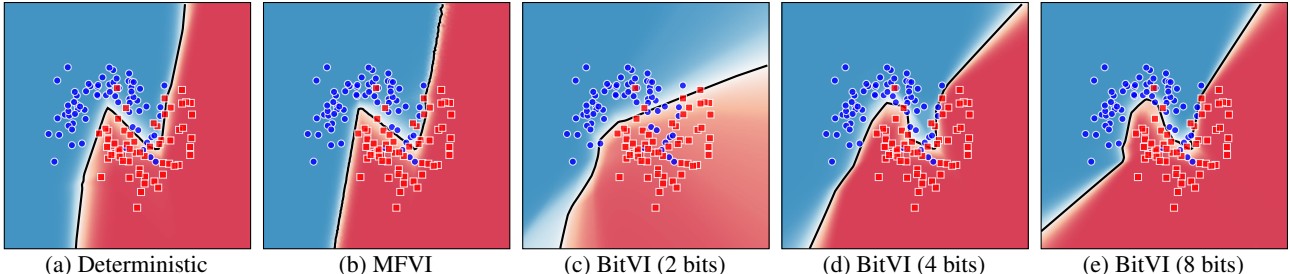

| (a) Deterministic | (b) MFVI | (c) BitVI (2 bits) | (d) BitVI (4 bits) | (e) BitVI (8 bits) |

Figure 6: **Uncertainty quantification in neural networks:** We consider the two moons binary ● classification problem with an MLP neural network (two hidden layers). The predictive density ( ) shows that BitVI provides both representative uncertainties and good decision boundaries compared to the deterministic and MFVI baselines.

For a given input $y$, we can compute the inverse CDF transform of $y$ under $q_{\boldsymbol{\theta}}$ using a series of linear transformations. In particular, for sum nodes $\mathsf{S}$ compute

$$F_{\mathsf{S}_\epsilon}^{-1}(y) = \begin{cases} F_{\mathsf{C}_{\epsilon 1}}^{-1}\left(\frac{y - w_{\epsilon 0}}{w_{\epsilon 1}}\right) & \text{if } y > w_{\epsilon 0} \\ F_{\mathsf{C}_{\epsilon 0}}^{-1}\left(\frac{y}{w_{\epsilon 0}}\right) & \text{otherwise} \end{cases}, \quad (6)$$

where $\mathsf{C}$ denotes a child node of $\mathsf{S}$, *i.e.* a sum or leaf node, and $\epsilon \in \bigcup_{j=0}^{B}\{0,1\}^j$ is a bitstring. If $\mathsf{C}$ is a leaf node, we compute the inverse CDF according to the respective leaf distribution, *i.e.*, $F_\psi^{-1}(y) = y(b-a) + a$ in case of a continuous uniform distribution $\mathsf{Unif}(a, b)$.

Note that the resulting value still requires discretization, and in case of fixed-point numbers needs to be rounded to the nearest fixed-point value. In fact, the bitstring $\epsilon$ generated by traversing the circuit in order to compute its inverse CDF already encodes the nearest fixed-point value for $y$. However, as the discretization operation does not have a well-defined gradient, we resort to the application of the straight-through estimator (STE) [Bengio et al., 2013]. In particular, we compute:

$$x = (\phi(\epsilon) + F_{q_{\boldsymbol{\theta}}}^{-1}(y)) - F_{q_{\boldsymbol{\theta}}}^{-1}(y), \quad (7)$$

where $\phi(\epsilon)$ is the mapping function defined by the number system and the bitstring $\epsilon$ is a function of $F_{q_{\boldsymbol{\theta}}}^{-1}$ and indicates the decision taken in Eq. (8).

**Representing Multivariate Distributions** So far, our induced variational distribution is only defined on the real line (univariate case). To extend the approach to the multivariate case, we considered two approaches: *(i)* a mean-field variational family, and *(ii)* a variational family model with dependencies between dimensions. To represent dependencies between the dimensions, we construct a deterministic PC representing the joint distribution over the bits of all the dimensions. In the case of fixed-point number systems, the resulting circuit model recursively splits the domain into hyper-rectangles by performing axis-aligned splits that alternate between dimensions in the construction. Note that this construction results in a binary tree consisting of $2^{B*D}$

leaves, where $B$ is the number of bits and $D$ is the number of dimensions. Thus, making it useful in low-dimensional or low-precision settings. However, including conditional independencies in the model can result in substantially more compact representations [Peharz et al., 2020, Garg et al., 2024]. We provide further details on the construction in Appendix A.

Applying the inverse CDF reparameterization for multivariate densities modeled with BitVI requires further considerations. In the case of the mean-field approximation, we apply the inverse CDF reparameterization (described above) independently for each dimension. If BitVI represents a variational distribution that models dependencies between dimensions, we employ the inverse of the tree-CDF transformation [Awaya and Ma, 2024], which is a map $\mathbb{R}^D \to [0, 1]^D$ where $D$ is the number of dimensions. In particular, for a given input $\boldsymbol{y} \in [0, 1]^D$, we compute the inverse tree-CDF transform of $\boldsymbol{y}$ by applying the following axis-aligned linear transformations at each sum node, where $\mathsf{S}_{d,\epsilon_d}$ denotes the sum node for dimension $d \leq D$ under bitstring $\epsilon_d$. The axis-aligned transformations are given as:

$$F_{\mathsf{S}_{d,\epsilon_d}}^{-1}(y_d) = \begin{cases} F_{\mathsf{C}_1}^{-1}\left(\frac{y_d - w_{d,\epsilon_d 0}}{w_{d,\epsilon_d 1}}\right) & \text{if } y_d > w_{d,\epsilon_d 0} \\ F_{\mathsf{C}_0}^{-1}\left(\frac{y_d}{w_{d,\epsilon_d 0}}\right) & \text{otherwise} \end{cases}, \quad (8)$$

where with some abuse of notation $\mathsf{C}_0$ denotes the left child of $\mathsf{S}_{d,\epsilon_d}$, which corresponds to a bit value of zero, and $\mathsf{C}_1$ denotes the right child (bit value of one). As we alternate dimensions at each level in the tree, decisions are made only based on the 'selected' dimension at each step. Computing the inverse of the tree-CDF transformation can still be performed efficiently, *i.e.*, in $\mathcal{O}(B * D)$ for $B$ bits.

## 4 EXPERIMENTS

Our experiments are designed to systematically validate the effectiveness of BitVI in performing approximate probabilistic inference over bitstring representations. In Section 4.1, we begin with 2D density estimation to demonstrate the expressiveness of our method in capturing complex non-

Table 1: **Bayesian benchmarks:** Negative log predictive density (NLPD±std, smaller better) results on the *Bayesian Benchmarks* UCI tasks (5-fold CV). We compare BitVI to Gaussian MFVI and Full-covariance Gaussian VI (FCGVI) on small MLP NN models. The best-performing method for each task is bolded, and multiple methods are bolded based on a paired $t$-test ($p = 5\%$). We show that BitVI works well on all test cases and is not significantly different from the baselines in most cases, even in the very low-bit range.

| Dataset | $(n, d)$ | MFVI | FCGVI | 2-BitVI | 4-BitVI | 8-BitVI |
|---|---|---|---|---|---|---|
| FERTILITY | (100,10) | **0.379**±0.107 | 0.406±0.111 | 0.728±0.139 | **0.407**±0.109 | **0.406**±0.142 |
| PITTSBURG-BRIDGES-T-OR-D | (102,8) | **0.345**±0.168 | **0.347**±0.078 | **0.301**±0.064 | **0.352**±0.082 | **0.391**±0.068 |
| ACUTE-INFLAMMATION | (120,7) | **0.004**±0.001 | 0.021±0.009 | **0.006**±0.002 | **0.006**±0.002 | 0.684±0.031 |
| ACUTE-NEPHRITIS | (120,7) | **0.003**±0.001 | 0.014±0.003 | **0.002**±0.000 | **0.002**±0.002 | 0.051±0.016 |
| ECHOCARDIOGRAM | (131,11) | **0.446**±0.167 | **0.515**±0.151 | **0.524**±0.200 | **0.435**±0.095 | 0.660±0.132 |
| HEPATITIS | (155,20) | **0.438**±0.081 | **0.447**±0.116 | 0.620±0.246 | 0.694±0.279 | **0.427**±0.085 |
| PARKINSONS | (195,23) | 0.322±0.151 | **0.284**±0.109 | **0.253**±0.098 | **0.261**±0.064 | **0.289**±0.061 |
| BREAST-CANCER-WISC-PROG | (198,34) | **0.540**±0.106 | **0.522**±0.128 | 0.699±0.087 | **0.584**±0.073 | **0.548**±0.087 |
| SPECT | (265,23) | **0.614**±0.067 | **0.624**±0.053 | 0.801±0.108 | 0.807±0.148 | **0.670**±0.125 |
| STATLOG-HEART | (270,14) | **0.478**±0.133 | **0.488**±0.156 | **0.550**±0.207 | 0.606±0.270 | **0.478**±0.147 |
| HABERMAN-SURVIVAL | (306,4) | **0.535**±0.062 | **0.523**±0.054 | **0.531**±0.042 | **0.525**±0.044 | **0.530**±0.036 |
| IONOSPHERE | (351,34) | **0.288**±0.094 | **0.276**±0.092 | 0.335±0.126 | **0.459**±0.217 | **0.323**±0.127 |
| HORSE-COLIC | (368,26) | **0.611**±0.159 | **0.595**±0.163 | 0.618±0.119 | 0.690±0.143 | **0.576**±0.103 |
| CONGRESSIONAL-VOTING | (435,17) | **0.670**±0.093 | **0.700**±0.126 | **0.699**±0.105 | **0.704**±0.108 | **0.644**±0.048 |
| CYLINDER-BANDS | (512,36) | **0.602**±0.107 | **0.633**±0.050 | 0.835±0.222 | **0.955**±0.361 | **0.678**±0.019 |
| BREAST-CANCER-WISC-DIAG | (569,31) | **0.078**±0.050 | **0.108**±0.029 | 0.148±0.080 | 0.172±0.152 | 0.155±0.097 |
| ILPD-INDIAN-LIVER | (583,10) | **0.547**±0.059 | **0.547**±0.033 | **0.535**±0.053 | **0.518**±0.032 | **0.567**±0.025 |
| MONKS-2 | (601,7) | **0.083**±0.121 | 0.607±0.082 | 0.563±0.060 | 0.656±0.073 | 0.666±0.030 |
| CREDIT-APPROVAL | (690,16) | **0.357**±0.025 | 0.417±0.096 | 0.405±0.041 | **0.358**±0.026 | **0.343**±0.009 |
| STATLOG-AUSTRALIAN-CREDIT | (690,15) | **0.662**±0.035 | 0.650±0.029 | 0.764±0.075 | **0.629**±0.019 | **0.626**±0.019 |
| BREAST-CANCER-WISC | (699,10) | **0.091**±0.042 | **0.105**±0.041 | **0.171**±0.113 | 0.168±0.055 | **0.122**±0.059 |
| BLOOD | (748,5) | **0.483**±0.058 | **0.483**±0.036 | **0.486**±0.057 | **0.478**±0.043 | **0.486**±0.039 |
| PIMA | (768,9) | 0.516±0.045 | **0.507**±0.042 | **0.512**±0.039 | **0.492**±0.031 | **0.492**±0.042 |
| MAMMOGRAPHIC | (961,6) | **0.428**±0.039 | 0.468±0.044 | **0.430**±0.053 | **0.417**±0.039 | **0.423**±0.049 |
| STATLOG-GERMAN-CREDIT | (1000,25) | **0.547**±0.066 | **0.557**±0.086 | 0.651±0.092 | **0.646**±0.101 | 0.894±0.249 |

Gaussian distributions. In Section 4.2, we explore Bayesian deep learning applications by applying BitVI to MLP neural networks, showcasing its ability to perform effective uncertainty quantification in predictive modeling. We then conduct a series of ablation studies in Section 4.3 to assess the trade-offs between numerical precision and model expressivity, investigating the effect of bitstring depth on performance and the role of hierarchical structure in neural networks.

**Implementation** The method was implemented in Python using the PyTorch library in order to facilitate automatic differentiation, convenient construction of neural network architectures, and fast parallelized training on GPUs. The training was conducted on a high-performance computing cluster with Nvidia [H,A,V,P]100, K80, and H200 GPUs. As a ballpark, the model training run time for single models in the experiments is measured in the range of minutes for the size of models we consider in these experiments.

### 4.1 2D DENSITIES

First, we demonstrate the flexibility of our proposed approach in 2D non-Gaussian target distributions. In Fig. 2, we include typical benchmark target densities (mixture, Neal's funnel, two-modal Gaussian, ring, and banana) that we approximate with 4-bit BitVI . Moreover, Fig. 7 shows a comparison for two densities, indicating that BitVI captures the overall density and cross-dependencies well, with approximation quality increasing with the number of bits. Fig. 11 in

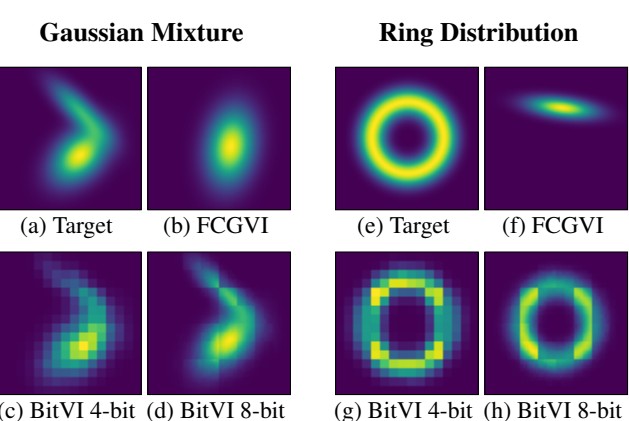

**Gaussian Mixture**
(a) Target (b) FCGVI
(c) BitVI 4-bit (d) BitVI 8-bit

**Ring Distribution**
(e) Target (f) FCGVI
(g) BitVI 4-bit (h) BitVI 8-bit

Figure 7: Comparison of 4-bit/8-bit BitVI against full-covariance Gaussian VI (FCGVI) on 2D non-Gaussian target distributions. A full comparison on all target distributions is given in Fig. 11. BitVI captures the overall density and cross-dependencies better than FCGVI.

the Appendix shows comparisons to the remaining densities.

### 4.2 MLP NEURAL NETWORK MODELS

We experiment with probabilistic inference in multi-layer perceptron (MLP) neural network (NN) models. For simplicity, we use similar neural network architectures in all the NN experiments. We use two hidden layers in all experiments, only varying the number of units. Additionally, we use the layer norm to ensure weight scaling.

Fig. 6 shows an uncertainty quantification example. We consider the two moons binary classification problem with an MLP neural network ([8,8] hidden units). The predictive density shows that BitVI provides both representative uncertainties and good decision boundaries compared to the deterministic and mean-field Gaussian VI baselines.

To give a more quantitative treatment to MLP NN modeling tasks, we use the *Bayesian Benchmarks*[1] community suite meant for benchmarking Bayesian methods in machine learning. Bayesian benchmarks include common evaluation data sets (typically from UCI [Kelly et al., 2025]) and make it possible to run a large number of comparisons under a fixed evaluation setup. We evaluate our approach in binary classification, and for an interesting probabilistic treatment, we include small-data binary classification tasks with $100 \leq n \leq 1000$ data samples (25 data sets). We follow the standard setup of input point normalization and splits in the evaluation suite. Additional details on the NN architectures and evaluation setup can be found in Appendix C.2.

Table 1 shows the results for BitVI (with 2, 4, and 8 bits), mean-field Gaussian VI (MFVI), and full-covariance Gaussian VI (FCGVI). Our approach consistently performs competitively with the standard variational inference baselines, even in the low-bit regime. Notably, in most data sets, BitVI with 4-bit and 8-bit representations achieves comparable performance to MFVI and FCGVI, demonstrating that probabilistic inference can be effectively conducted over bitstring representations without significant loss in predictive power. Even at 2-bit precision, BitVI remains viable in several cases. Yet, the results also suggest that more flexible probabilistic modeling in this neural network setting might not be needed, as the 8-bit models show very little or any benefits over the 4-bit models.

### 4.3 ABLATION STUDIES

**Increasing Complexity of Target Distribution** We consider an ablation study where we control the target distribution complexity. For this, we constructed a mixture of equidistant Gaussians and assessed the entropy of BitVI under varying numbers of bits under three different amounts of variance for each Gaussian. Fig. 9 shows the fitted results of BitVI (black) with 16 bits for target distributions with increasing complexity (gray) alongside the entropy of BitVI under varying number of bits. The entropy (lower figures) shows the cut-off for number of bits needed to represent each target, indicating that BitVI naturally exhibits a parsimonious behaviour.

**Trade-off Between Model Complexity and Bitstring Depth** For NN applications, an interesting question is

---

[1]github.com/secondmind-labs/bayesian_benchmarks; originally by Salimbeni *et al.*

Table 2: The trade-off between NN model complexity (units in hidden layers) and bitstring length (2–12 bits). The negative log predictive density (NLPD, smaller better) on the two moons data suggests that even low bit depth models perform well, and the dominating factor in expressivity is the number of units in the NN. See Appendix D for ACC/ECE.

| | | Increasing NN complexity $\rightarrow$ | | | | | |
|---|---|---|---|---|---|---|---|
| | | [4, 4] | [6, 6] | [8, 8] | [10, 10] | [12, 12] | [14, 14] | [16, 16] |
| | 2 | 0.36 | 0.35 | 0.35 | 0.32 | 0.33 | 0.3 | 0.29 |
| | 3 | 0.37 | 0.36 | 0.26 | 0.34 | 0.27 | 0.24 | 0.25 |
| | 4 | 0.38 | 0.32 | 0.31 | 0.3 | 0.27 | 0.28 | 0.24 |
| | 5 | 0.35 | 0.32 | 0.36 | 0.29 | 0.27 | 0.25 | 0.25 |
| Bitstring depth | 6 | 0.34 | 0.34 | 0.37 | 0.3 | 0.28 | 0.25 | 0.24 |
| | 7 | 0.31 | 0.3 | 0.3 | 0.26 | 0.28 | 0.25 | 0.24 |
| | 8 | 0.33 | 0.31 | 0.25 | 0.3 | 0.29 | 0.26 | 0.26 |
| | 9 | 0.36 | 0.32 | 0.32 | 0.33 | 0.26 | 0.23 | 0.25 |
| | 10 | 0.33 | 0.35 | 0.3 | 0.3 | 0.25 | 0.26 | 0.24 |
| | 12 | 0.37 | 0.29 | 0.35 | 0.35 | 0.26 | 0.27 | 0.24 |

whether fine-grained numerical accuracy is needed to represent the model weights in the first place. Recent advances in large-scale model training and inference suggest that rather than numerical accuracy, the models benefit from more parameters, which enable further flexibility. Hence, we study whether the models benefit from higher numerical granularity w.r.t. probabilistic treatment.

In Table 2, we vary both the neural network complexity (units in the two hidden layers) and the bitstring length. We consider 2–12-bit models (with only fractional bits). The negative log predictive density (NLPD, smaller better) on the two moons data suggests that even low bit depth models perform well, and the dominating factor in expressivity is the number of units in the NN. In Appendix D, we include similar tables for both accuracy and expected calibration error (ECE).

**Do Bitstrings Capture Hierarchies in NNs?** Finally, we use a neural network model to study the hierarchies captured by BitVI. We start from a 10-bit NN BitVI results on the Banana binary classification data set and gradually decrease the fractional precision of the trained model, chopping off more granular levels of the model. Fig. 8 shows the results for 10, 8, 6, 4, and 2-bit models (2 integer bits each, except for the 2-bit model). Even the 4-bit model (2 integer bits and 1 fractional bit) captures the overall structure well, whereas the 2-bit model (with no integer bits; only a sign bit and a fraction bit) struggles.

## 5 DISCUSSION AND CONCLUSION

In this work, we introduced **BitVI**, a novel approach for approximate Bayesian inference that operates directly in the space of discrete bitstring representations. By leveraging (deterministic) probabilistic circuits as the representational framework, we demonstrated that inference can be performed directly on bitstring representations of number

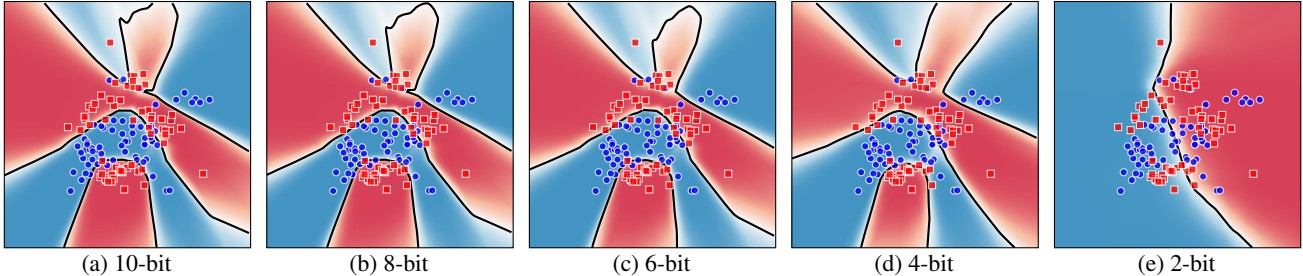

| (a) 10-bit | (b) 8-bit | (c) 6-bit | (d) 4-bit | (e) 2-bit |

Figure 8: **Chopping the banana:** We start from a 10-bit NN BitVI results on the Banana binary classification data set and gradually decrease the fractional precision of the trained model. The low-bit models up to 4 bits capture the overall structure well. This is further confirmed by the results in Table 2.

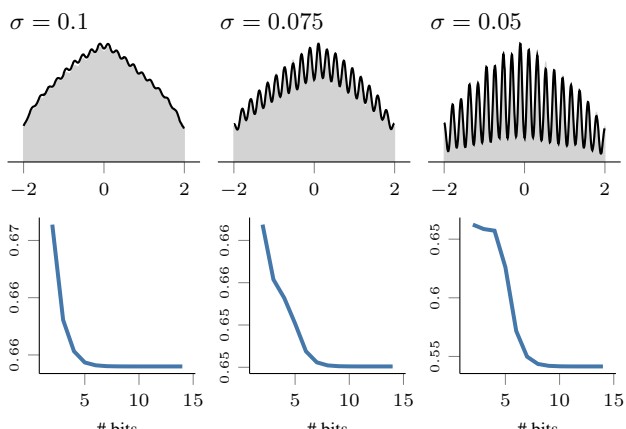

Figure 9: Ablation result of BitVI (black) for target distributions with increasing complexity (gray) and the precision used by the variational distribution to represent the target. The entropy (lower figures) shows the cut-off for bitstring depth needed to represent each target.

systems, enabling effective approximate inference and uncertainty quantification. Our approach presents a paradigm shift by learning a rich variational approximation induced by a variational family on bitstring representations without relying on high-precision representations. Our experiments showcased the flexibility of BitVI across different settings: In Section 4.1, we illustrated its ability to approximate complex non-Gaussian densities; and in Section 4.2, we demonstrated its effectiveness in Bayesian deep learning, where it provided robust uncertainty estimates while maintaining computational efficiency.

Beyond demonstrating feasibility, these results highlight that flexible approximate Bayesian inference does not need to be constrained to continuous-valued computations but can be reformulated in a fully discrete manner. Moreover, our results further highlight the potential of using probabilistic circuits as the representational framework for approximate inference. While BitVI provides a promising direction for flexible variational inference, several limitations remain, which we will briefly discuss.

**Limitations**   In order to scale to high-dimensional settings, our approach currently needs to employ a mean-field approximation to the posterior. This limitation arises from our tree construction, which considers dependencies between all bits and dependencies between all dimensions if no mean-field assumption is made. In practical applications, modeling all dependencies is likely unnecessary and introduces an excessive computational and memory burden. Therefore, a promising future direction is to leverage more compact representations such [Peharz et al., 2020]. For the same reason, our approach currently introduces many parameters to be optimized, which can result in a challenge for high-dimensional settings. Lastly, our experiments are currently limited to fixed-point representations, and exploiting the representational power of floating-point representations would be a promising future avenue.

The codes and resources for BitVI will be made available on GitHub upon publication of the paper[2].

## Acknowledgements

A. Solin acknowledges funding from the Research Council of Finland (grant number 339730). M. Trapp acknowledges funding from the Research Council of Finland (grant number 347279). A. Sladek acknowledges funding from the Finnish Doctoral Program Network in Artificial Intelligence (AI-DOC, decision number VN/3137/2024-OKM-6). We acknowledge the computational resources provided by the Aalto Science-IT project. We thank the reviewers and the area chair for their constructive feedback.

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

# Approximate Bayesian Inference via Bitstring Representations (Supplementary Material)

**Aleksanteri Sladek**[1]          **Martin Trapp**[1]          **Arno Solin**[1]

[1]Department of Computer Science, Aalto University, Espoo, Finland

## A   TECHNICAL DETAILS

### A.1   PROBABILISTIC CIRCUITS

We will briefly review the main concepts related to probabilistic circuits (PC), relevant for this work.

**Definition A.1** (scope of a node). *The scope of a node is the set of variables it depends on. See [Trapp et al., 2019] for details.*

**Definition A.2** (smooth & decomposable circuit). *A sum node is smooth if its children have the same scope. A product node is decomposable if its children have pairwise disjoint scopes. A circuit is smooth (resp. decomposable) if all its sum nodes are smooth (resp. product nodes are decomposable).*

In this work, we only consider circuits that fullfil both smoothness and decomposability conditions as they both are required to render common inference tasks, such as density evaluation and marginalisation, tractable.

### A.2   MULTIVARIATE BITSTRING REPRESENTATIONS

As outlined in the main text, for multivariate distributions, we generate a circuit model that represents a distribution over hyper-rectangles. Let $\Omega$ denote the domain of the distribution, we recursively construct a dyadic partition of the domain into measurable subsets. This process is done by selecting a splitting dimension at each level of the tree and splitting the hyper-rectangle according to the number system representation, *i.e.*, in the middle for fixed-point numbers. At the next level, we select a splitting dimension our of the remaining dimension (those that have not been split yet) and split the hyper-rectangle accordingly. We make sure each dimension has been split in the process, before restarting the splitting. The construction ends if each dimension has been split $B$ many times, where $B$ is the number of bits used in the number system. Fig. 10 illustrates the recursive splitting of the input domain $\Omega$ into sub-domains (hyper-rectangles).

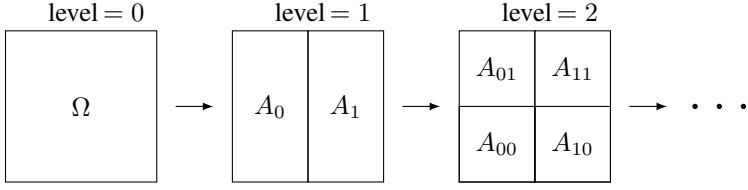

Figure 10: Illustration of the iterative axis-aligned splitting of the domain into hyper-rectangles (sub-domains) by the circuit.

## B DERIVATIONS

### B.1 ENTROPY CALCULATION EXAMPLE FOR A DETERMINISTIC PC

Let $\mathcal{C}$ be a deterministic PC with a sum node $\mathsf{S}_0$ as its root and two children $\mathsf{P}_{00}$ and $\mathsf{P}_{01}$. Product nodes have two children, one of which is a leaf node and the other a sum node. For example, $\mathsf{P}$ has a leaf node $\mathsf{L}_00$ and a sum node $\mathsf{S}_00$ as its children. Leaf nodes are indicator functions $\mathsf{L}_{00}(\mathbf{x}) = \mathbb{1}\{x_0 = 0\}$, where $\mathbb{1}\{\cdot\}$ is the indicator function. All sum nodes in the circuit have two children. Hence, $\mathsf{S}_0$ has weights $w_{00}$ and $1 - w_{00}$, where $0 \leq w_{00} \leq 1$.

$$\mathcal{H}\left[\mathcal{C}(\mathbf{x})\right] = -\int_{\mathbf{x} \in \mathcal{X}} \mathcal{C}(\mathbf{x}) \log \mathcal{C}(\mathbf{x}) \tag{9}$$

$$= -\int_{\mathbf{x} \in \mathcal{X}} \mathsf{S}_0(\mathbf{x}) \log\left(\mathsf{S}_0(\mathbf{x})\right) \tag{10}$$

$$= -\int_{\mathbf{x} \in \mathcal{X}} \left[w_{00}\mathsf{P}_{01}(\mathbf{x}) + (1 - w_{00})\mathsf{P}_{00}(\mathbf{x})\right] \log\left\{\left[w_{00}\mathsf{P}_{01}(\mathbf{x}) + (1 - w_{00})\mathsf{P}_{00}(\mathbf{x})\right]\right\} \tag{11}$$

$$= -\int_{\mathbf{x} \in \mathcal{X}} \left[w_{00}\mathsf{L}_{01}(x_0) * \mathsf{S}_{00}(\hat{\mathbf{x}}) + (1 - w_{00})\mathsf{L}_{01}(x_0) * \mathsf{S}_{00}(\hat{\mathbf{x}})\right] * \tag{12}$$

$$\log\left\{\left[w_{0,0}\mathsf{L}_{01}(x_0) * \mathsf{S}_{0,0}(\hat{\mathbf{x}}) + (1 - w_{00})\mathsf{L}_{00}(x_0)\mathsf{S}_{0,1}(\hat{\mathbf{x}})\right]\right\} \tag{13}$$

$$= -\int_{\mathbf{x} \in \mathcal{X}} \left[w_{00}\mathbb{1}\{x_0 = 1\}\mathsf{S}_{01}(\hat{\mathbf{x}}) + (1 - w_{00})\mathbb{1}\{x_0 = 0\}\mathsf{S}_{00}(\hat{\mathbf{x}})\right] * \tag{14}$$

$$\log\left\{\left[w_{00}\mathbb{1}\{x_0 = 1\}\mathsf{S}_{01}(\hat{\mathbf{x}}) + (1 - w_{00})\mathbb{1}\{x_0 = 0\}\mathsf{S}_{00}(\hat{\mathbf{x}})\right]\right\}. \tag{15}$$

Where $\hat{\mathbf{x}}$ denotes the vector $\mathbf{x}$ without variable $x_0$.

Next, partition the integral into two integrals leveraging the fact that,

$$\int_{x \in \mathcal{X}} f(x)dx = \int_{x \in \mathcal{X}_A} f(x)\mathrm{d}x + \int_{x \in \mathcal{X}_B} f(x)\mathrm{d}x, \ \mathcal{X} = \mathcal{X}_A \bigcup \mathcal{X}_B, \mathcal{X}_A \bigcap \mathcal{X}_B = \emptyset. \tag{16}$$

Here, the integral splits into subsets $\mathcal{X}_{x_0}$ representing the set of all bit vectors $\boldsymbol{x}$ with $x_0 = 1$, and respectively $\mathcal{X}_{\neg x_0}$ with all bit vectors $\boldsymbol{x}$ with $x_0 = 0$. Hence,

$$\mathcal{H}[\mathcal{C}(\boldsymbol{x})] = -\int_{\mathbf{x} \in \mathcal{X}_{x_0}} \left[w_{00}\mathbb{1}\{x_0 = 1\}\mathsf{S}_{01}(\mathbf{x}) + (1 - w_{00})\mathbb{1}\{x_0 = 0\}\mathsf{S}_{00}(\mathbf{x})\right] * \tag{17}$$

$$\log\left\{\left[w_{00}\mathbb{1}\{x_0 = 1\}\mathsf{S}_{01}(\mathbf{x}) + (1 - w_{00})\mathbb{1}\{x_0 = 0\}\mathsf{S}_{00}(\mathbf{x})\right]\right\} \tag{18}$$

$$- \int_{\mathbf{x} \in \mathcal{X}_{\neg x_0}} \left[w_{00}\mathbb{1}\{x_0 = 1\}\mathsf{S}_{01}(\mathbf{x}) + (1 - w_{00})\mathbb{1}\{x_0 = 0\}\mathsf{S}_{00}(\mathbf{x})\right] * \tag{19}$$

$$\log\left\{\left[w_{00}\mathbb{1}\{x_0 = 1\}\mathsf{S}_{01}(\mathbf{x}) + (1 - w_{00})\mathbb{1}\{x_0 = 0\}\mathsf{S}_{00}(\mathbf{x})\right]\right\}. \tag{20}$$

Now, each integral can be simplified since the indicator functions will always evaluate to 0 or 1 in the respective subsets of $\mathcal{X}$, i.e.,

$$\mathcal{H}[\mathcal{C}(\boldsymbol{x})] = -\int_{\mathbf{x} \in \mathcal{X}_{x_0}} w_{00}\mathsf{S}_{01}(\mathbf{x}) \log\left\{\left[w_{00}\mathsf{S}_{01}(\mathbf{x})\right]\right\} \tag{21}$$

$$- \int_{\mathbf{x} \in \mathcal{X}_{\neg x_0}} \left[(1 - w_{00})\mathsf{S}_{00}(\mathbf{x})\right] * \log\left\{(1 - w_{00})\mathsf{S}_{00}(\mathbf{x})\right\}. \tag{22}$$

As the two integrals have the same form, for notational simplicity, we will only consider the first integral (in orange). The

second integral can be computed in the same way.

$$-\int_{\mathbf{x}\in\mathcal{X}_{x_0}} w_{00}\mathsf{S}_{01}(\mathbf{x})\log\left\{[w_{00}\mathsf{S}_{01}(\mathbf{x})]\right\} \tag{23}$$

$$= -\int_{\mathbf{x}\in\mathcal{X}_{x_0}} w_{00}\mathsf{S}_{01}(\mathbf{x})\left[\log w_{00} + \log\left(\mathsf{S}_{01}(\mathbf{x})\right)\right] \tag{24}$$

$$= -\int_{\mathbf{x}\in\mathcal{X}_{x_0}} w_{00}\log(w_{00})\mathsf{S}_{01}(\mathbf{x}) + w_{00}\mathsf{S}_{01}(\mathbf{x}) * \log\left(\mathsf{S}_{01}(\mathbf{x})\right) \tag{25}$$

$$= -w_{00}\log(w_{00})\int_{\mathbf{x}\in\mathcal{X}_{x_0}} \mathsf{S}_{01}(\mathbf{x}) - w_{00}\int_{\mathbf{x}\in\mathcal{X}_{x_0}} \mathsf{S}_{01}(\mathbf{x}) * \log\left(\mathsf{S}_{01}(\mathbf{x})\right). \tag{26}$$

Notice that the second integral is the entropy of the sum node $\mathsf{S}_{01}$. Hence,

$$= -w_{00}\log(w_{00})\int_{\mathbf{x}\in\mathcal{X}_{x_0}} \mathsf{S}_{01}(\hat{\mathbf{x}}) + w_{00}\mathcal{H}(\mathsf{S}_{01}(\hat{\boldsymbol{x}})). \tag{27}$$

Furthermore, if $\mathsf{S}_{01}$ is normalized, then $\int_{\mathbf{x}\in\mathcal{X}_{x_0}} \mathsf{S}_{01}(\mathbf{x})\mathrm{d}\boldsymbol{x} = 1$, leading to the further simplification,

$$= -w_{00}\log(w_{00}) + w_{00}\mathcal{H}(\mathsf{S}_{01}(\hat{\boldsymbol{x}})). \tag{28}$$

### B.2 REVERSE KL DIVERGENCE CALCULATION

Let us define a density $q$ and a density $p$. The reverse KL divergence of $q$ from $p$ is denoted as $\mathrm{KL}(q\,\|\,p)$, and defined as:

$$\mathrm{KL}(q\,\|\,P) = \int q(\boldsymbol{x})\log\frac{p(\boldsymbol{x})}{q(\boldsymbol{x})}\mathrm{d}\boldsymbol{x} \tag{29}$$

$$= -\int q(\boldsymbol{x})\log q(\boldsymbol{x})\mathrm{d}\boldsymbol{x} + \int q(\boldsymbol{x})\log p(\boldsymbol{x})\,\mathrm{d}\boldsymbol{x}. \tag{30}$$

Note that $-\int q(\boldsymbol{x})\log q(\boldsymbol{x})\,\mathrm{d}\boldsymbol{x}$ is the entropy of distribution $q$, and will be denoted as $-\mathcal{H}(q)$:

$$\mathrm{KL}(q\,\|\,p) = -\int q(\boldsymbol{x})\log p(\boldsymbol{x})\,\mathrm{d}\boldsymbol{x} - \mathcal{H}(q). \tag{31}$$

Note also that $\int q(\boldsymbol{x})\log p(\boldsymbol{x})\,\mathrm{d}$ is the expected value of the log-likelihood of $p$ w.r.t. $q$:

$$\mathrm{KL}(q\,\|\,p) = -\mathbb{E}_{\boldsymbol{x}\sim q}\left[\log p(\boldsymbol{x})\right] - \mathcal{H}(q). \tag{32}$$

## C EXPERIMENTAL DETAILS

### C.1 2D DENSITIES

We present results for 2D non-Gaussian target distributions. In Fig. 11, we include additional results for typical benchmark target densities (mixture, Neal's funnel, two-modal Gaussian, ring, and banana) that we approximate with 4-bit/8-bit BitVI, which captures the overall density and cross-dependencies well.

### C.2 MLP NEURAL NETWORK MODELS

The experiments with the *Bayesian-benchmarks* data sets used the following hyperparameters and setup:

- Adam optimizer with a learning rate of 0.001
- Hidden layer size $16\times16$ for $D \leq 500$ and $32\times32$ for $D > 500$
- Batch size of 32 for $D \leq 500$ and 128 for $D > 500$

- 64 samples for computing the Monte Carlo approximation of the posterior log-joint
- Weight representations used two integer bits, except for the 2-bit model, which used zero integer bits
- LayerNorm [Ba et al., 2016] applied to hidden layers (pre-activation)
- Depth-based regularization for circuit parameters $\epsilon\, d^2$ with $\epsilon = 0.1$
- Early stopping based on the validation set ELBO loss after 2000 epochs
- Circuit weights were initialized from a beta distribution based on the height of the sum node in the circuit. The beta distribution $\alpha$ and $\beta$ were set as $2^h$ where $h$ is the height of the sum node in the circuit.
- 5-fold cross-validation into train and test sets
- Validation set split from the train set with 20% of the train set data

## C.3   ABLATION STUDIES

**Banana Chopping**

- Training set of 2048 points
- Validation set of 512 points
- Adam optimizer with a learning rate of 0.01
- Batch size of 256
- LayerNorm [Ba et al., 2016] applied to hidden layers (pre-activation)
- Weight representations used 10 bits with no integer bits. A sign bit and nine fractional bits.
- Depth-based regularization for circuit parameters $\epsilon\, d^2$ with $\epsilon = 0.001$.
- Circuit weights were initialized from a beta distribution based on the height of the sum node in the circuit. The beta distribution $\alpha$ and $\beta$ were set as $2^h$ where $h$ is the height of the sum node in the circuit.

# D   ADDITIONAL RESULTS

The following section contains additional results.

## D.1   ABLATION STUDIES

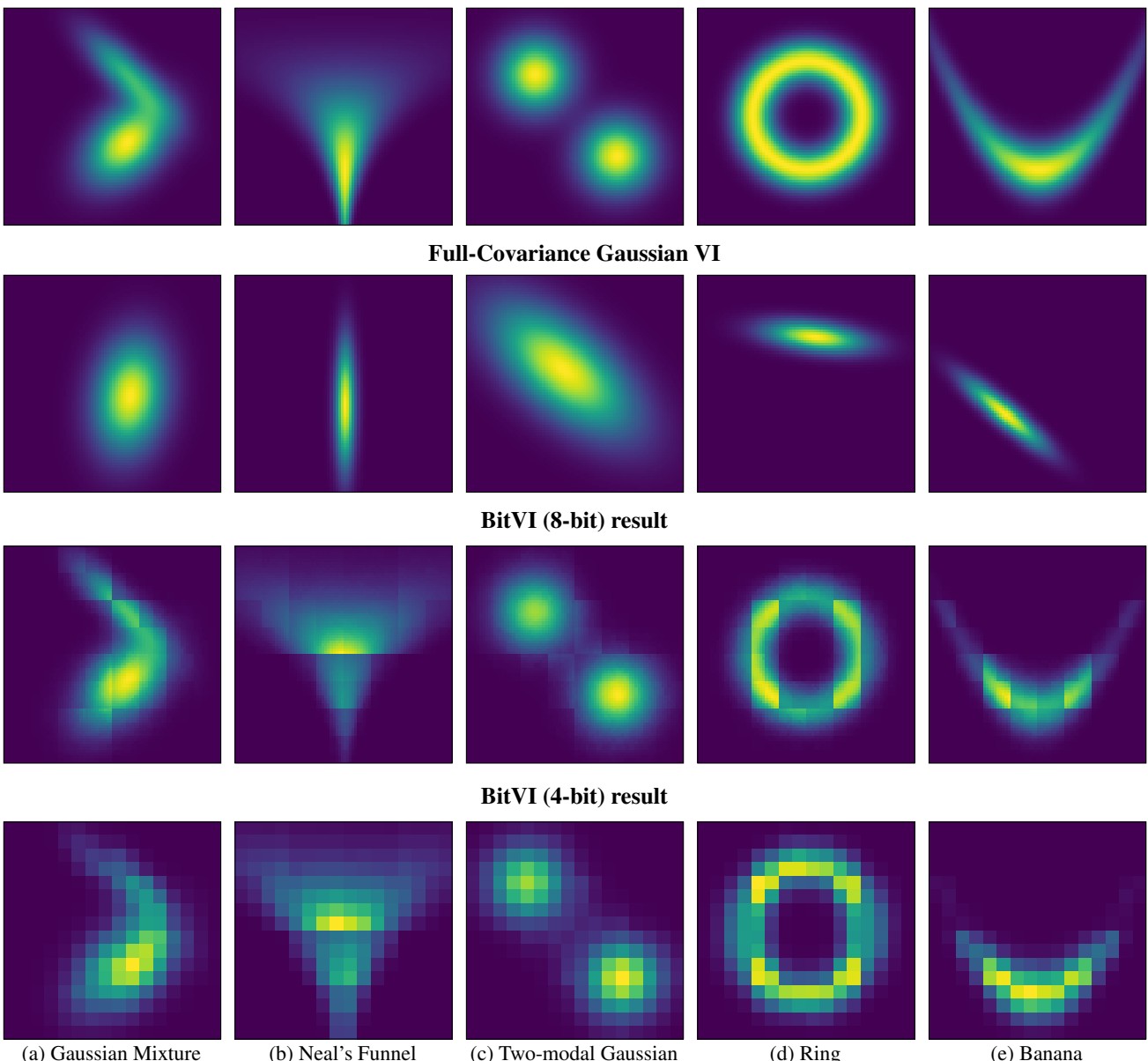

Figure 11: 2D non-Gaussian target distributions. We include results for typical benchmark target densities (mixture, Neal's funnel, two-modal Gaussian, ring, and banana) that we approximate with 4-bit/8-bit BitVI, which captures the overall density and cross-dependencies well.

Table 3: Trade-off between NN model complexity (units in hidden layers) and bitstring depth (2–12 bits). Accuracy and expected calibration error (ECE) on the two moons data suggest that even low bit depth models perform well, and the dominating factor in expressivity is the number of units in the NN. See Table 2 in the main paper for the NLPD.

(a) Accuracy

|    | [4, 4] | [6, 6] | [8, 8] | [10, 10] | [12, 12] | [14, 14] | [16, 16] |
|----|--------|--------|--------|----------|----------|----------|----------|
| 2  | 0.856  | 0.85   | 0.852  | 0.87     | 0.868    | 0.888    | 0.89     |
| 3  | 0.854  | 0.857  | 0.903  | 0.865    | 0.897    | 0.909    | 0.906    |
| 4  | 0.852  | 0.879  | 0.88   | 0.884    | 0.904    | 0.898    | 0.909    |
| 5  | 0.86   | 0.877  | 0.854  | 0.895    | 0.901    | 0.909    | 0.913    |
| 6  | 0.863  | 0.861  | 0.853  | 0.887    | 0.895    | 0.91     | 0.906    |
| 7  | 0.882  | 0.883  | 0.888  | 0.899    | 0.896    | 0.909    | 0.905    |
| 8  | 0.874  | 0.877  | 0.909  | 0.884    | 0.886    | 0.909    | 0.904    |
| 9  | 0.864  | 0.873  | 0.873  | 0.877    | 0.897    | 0.914    | 0.909    |
| 10 | 0.87   | 0.862  | 0.886  | 0.884    | 0.912    | 0.899    | 0.909    |
| 12 | 0.852  | 0.888  | 0.863  | 0.863    | 0.898    | 0.895    | 0.908    |

(b) ECE

|    | [4, 4] | [6, 6] | [8, 8] | [10, 10] | [12, 12] | [14, 14] | [16, 16] |
|----|--------|--------|--------|----------|----------|----------|----------|
| 2  | 0.059  | 0.053  | 0.061  | 0.064    | 0.065    | 0.055    | 0.06     |
| 3  | 0.06   | 0.071  | 0.051  | 0.053    | 0.046    | 0.042    | 0.049    |
| 4  | 0.064  | 0.064  | 0.055  | 0.045    | 0.042    | 0.045    | 0.038    |
| 5  | 0.061  | 0.057  | 0.058  | 0.053    | 0.042    | 0.043    | 0.044    |
| 6  | 0.06   | 0.067  | 0.057  | 0.048    | 0.044    | 0.039    | 0.046    |
| 7  | 0.063  | 0.053  | 0.048  | 0.046    | 0.041    | 0.042    | 0.046    |
| 8  | 0.061  | 0.051  | 0.038  | 0.047    | 0.048    | 0.045    | 0.042    |
| 9  | 0.055  | 0.056  | 0.053  | 0.053    | 0.045    | 0.04     | 0.042    |
| 10 | 0.057  | 0.058  | 0.056  | 0.05     | 0.037    | 0.047    | 0.046    |
| 12 | 0.064  | 0.05   | 0.055  | 0.053    | 0.041    | 0.05     | 0.045    |