# OpenReview forum: "Approximate Bayesian Inference via Bitstring Representations"
_auai.org/UAI/2025/Workshop/TPM — TPM 2025_

### Official Review · Reviewer_CmX3 · 2025-06-15

**Rating:** 3

**Review:**

The authors consider a representation of a density over the real numbers based on fixed-point representations of real numbers, i.e., a bitstring.  One can represent a decision tree that maps bitstrings to their corresponding real numbers (at the leaves), and then induce a distribution on top of that (by associating weights the edges).  The authors propose to find these weights via variational inference (optimizing the ELBO).   The approach is also extended to 2D densities.  The approach is also evaluated on simulated 2D densities, and for learning Bayesian neural networks.

The approach is intuitive, and appears to work well, particularly for the 2D case.  I believe it would be of interest to TPM.

Other comments:

= For the 1D case, if one treated the decision tree like a Probabilistic Sentential Decision Diagram (PSDD), or more simply, if one just maximized the log likelihood, I believe one would derive a simple closed form for the weights.  For example, if your first bit in your fixed-point representation was the sign bit, the first weights on the root would simply count the number of positive versus the number of negative samples in your data.  More generally, each weight would then be the number of samples that used that bit, given the path taken to reach the node (and taking the corresponding subset of the data).

= I have hopefully a simple question about the formal definition.  A function phi is defined that maps bistrings to real numbers.  Shortly after, this function is inverted to define a distribution over real numbers.  However, one cannot map the all real numbers to a bitstring, since bitstrings are countable and real numbers are not countable.  So, unless I am missing something, this doesn't seem to be well-defined.  Later on, it is mentioned that the real numbers are basically mapped back into a fixed-point value, which is what I would expect.  On the other hand, in the examples in the figures, the leaves are represented using intervals of real numbers.  So, to me, these are basically three different things.

---

### Official Review · Reviewer_VVFZ · 2025-06-16
**interesting approach but a little bit confusing**

**Rating:** 2

**Review:**

While the paper presents an interesting approach to perform approximate inference in continuous domains the paper is a bit confusing. This is first and foremost the case for the neural network experiments. Here, the task is to perform binary classification using neural networks implemented with fixed point numbers. This is a bit strange as binary classification is usually trained with binary cross entropy. However, the main theoretical contribution is about computing the ELBO (generative model). It is not clear how those fit with the discriminative model for classification.
Another point is that much of the derivations in the appendix for the entropy of a deterministic arithmetic circuits as already present in the literature [1,2] without proper reference.
Lastly, the claims that the approach would scale to higher dimensions seem a bit overoptimistic (based on the content of the paper)

[1] work by Mario A. T. Figueiredo and colleaues, as well as work by Jason Eisner and colleagues

[2] Shih and Ermon, "Probabilistic Circuits for Variational Inference in Discrete Graphical Models"